# The Profound Influence of Gut Microbiome and Extracellular Vesicles on Animal Health and Disease

**DOI:** 10.3390/ijms25074024

**Published:** 2024-04-04

**Authors:** Muttiah Barathan, Sook Luan Ng, Yogeswaran Lokanathan, Min Hwei Ng, Jia Xian Law

**Affiliations:** 1Centre for Tissue Engineering and Regenerative Medicine, Faculty of Medicine, Universiti Kebangsaan Malaysia, Cheras, Kuala Lumpur 56000, Malaysia; lyoges@ppukm.ukm.edu.my (Y.L.); angela@ukm.edu.my (M.H.N.); 2Department of Craniofacial Diagnostics and Biosciences, Faculty of Dentistry, Universiti Kebangsaan Malaysia, Jalan Raja Muda Abdul Aziz, Kuala Lumpur 50300, Malaysia; ngsookluan@ukm.edu.my

**Keywords:** animal, gut microbiome, immune system regulation, dysbiosis, extracellular vesicles

## Abstract

The animal gut microbiota, comprising a diverse array of microorganisms, plays a pivotal role in shaping host health and physiology. This review explores the intricate dynamics of the gut microbiome in animals, focusing on its composition, function, and impact on host–microbe interactions. The composition of the intestinal microbiota in animals is influenced by the host ecology, including factors such as temperature, pH, oxygen levels, and nutrient availability, as well as genetic makeup, diet, habitat, stressors, and husbandry practices. Dysbiosis can lead to various gastrointestinal and immune-related issues in animals, impacting overall health and productivity. Extracellular vesicles (EVs), particularly exosomes derived from gut microbiota, play a crucial role in intercellular communication, influencing host health by transporting bioactive molecules across barriers like the intestinal and brain barriers. Dysregulation of the gut–brain axis has implications for various disorders in animals, highlighting the potential role of microbiota-derived EVs in disease progression. Therapeutic approaches to modulate gut microbiota, such as probiotics, prebiotics, microbial transplants, and phage therapy, offer promising strategies for enhancing animal health and performance. Studies investigating the effects of phage therapy on gut microbiota composition have shown promising results, with potential implications for improving animal health and food safety in poultry production systems. Understanding the complex interactions between host ecology, gut microbiota, and EVs provides valuable insights into the mechanisms underlying host–microbe interactions and their impact on animal health and productivity. Further research in this field is essential for developing effective therapeutic interventions and management strategies to promote gut health and overall well-being in animals.

## 1. The Gut Microbiome

The animal microbiota, consisting of a staggering nearly 100 trillion microorganisms encompassing bacteria, fungi, viruses, and small parasitic worms, is predominantly found in the gastrointestinal (GI) tract, also known as the gut [1]. The intestinal microbiota has long been recognized for its importance, particularly in the realm of veterinary medicine. These microorganisms colonize various anatomical sites of the animal host other than GI, such as the skin, mucosal surfaces (respiratory tract, urogenital tract), and internal organs, forming complex microbial communities known as microbiomes [2]. Early-life microbial succession in the gut of animals refers to the dynamic process by which diverse microbial communities, primarily bacteria, establish themselves in the gut of a newborn. It is a crucial stage that shapes the animal’s health and resilience throughout its life through gut maturation, whereby microbes stimulate the development of the gut’s lining, including the immune system and digestive processes [3,4].

Newborn health is significantly influenced by gut bacteria, with recent research suggesting a surprisingly active microbial environment in the womb. This challenges the traditional view of a sterile uterus and raises intriguing possibilities about how the initial microbiome is shaped [5]. This initial colonization phase is crucial for neonatal development, exemplified in ruminants where the colonization of the rumen microbiome during the birth-to-weaning period is pivotal and linked to the concept of coevolution between microorganisms and the host [6]. Early-life microbial succession, crucial for establishing the foundation of the gut microbiota, is influenced by various factors. The mode of delivery at birth shapes initial colonization, with vaginal delivery facilitating transmission of beneficial bacteria like Lactobacillus and Bifidobacterium [7]. Maternal microbiota, especially in the birth canal and breast milk, provides an inoculum rich in diverse microbes and bioactive compounds, fostering microbial diversity and immune development in infants [8]. Dietary practices, such as breastfeeding and introduction of complementary foods, further influence the gut microbiota composition. Antibiotic exposure, environmental factors like pet ownership, and host genetics also play significant roles [9,10]. Understanding these factors enables interventions to promote the establishment of a resilient and balanced gut microbiota early in life, with potential long-term impacts on health and disease susceptibility.

## 2. Function of Gut Microbiota

The gut microbes, known collectively as the microbiota, and often referred to as the “forgotten organ”, performs vital functions ranging from digestion and nutrient absorption to immune system regulation and metabolic homeostasis [11]. Through intricate interactions with the host and among themselves, these microorganisms form a dynamic and symbiotic relationship that profoundly influences various aspects of physiology, contributing beneficial effects on the host such as maintenance of gut health [12]. Commensal gut microbiota contributes to the development and maintenance of gut structure and morphology in healthy animals. They aid in the maturation of the intestinal epithelium, promote the growth of gut-associated lymphoid tissue (GALT), and help maintain the integrity of the gut barrier function, thereby preventing the translocation of pathogens into systemic circulation [13,14]. Gut microbiota plays a crucial role in modulating the host immune system. Specifically, an animal’s innate immune system employs an array of anatomical defenses to safeguard against microbial invasion [15,16]. These defenses include physical barriers such as the skin and mucosa, mechanical mechanisms like the expulsion of mucus and feces, and the microbiome, which consists of resident bacteria on the skin and in the gut [17]. Meanwhile, the adaptive immune system is a crucial layer of defense that provides protection against a wide range of microorganisms and can be broadly categorized into antibody (humoral) immunity, which targets extracellular invaders, and cell-mediated immunity, which targets intracellular invaders [18]. They stimulate the development and maturation of immune cells, such as T cells, B cells, and dendritic cells, in the GALT. This immune modulation helps in mounting appropriate immune responses to pathogens while preventing obesity, allergic diseases, inflammatory disorders, and autoimmune diseases, influencing an animal’s susceptibility to IgE-mediated immune reactions and allergies [19]. Commensal gut microbiota competes with pathogenic microorganisms for nutrients and adhesion sites in the gut, thereby inhibiting the colonization and proliferation of harmful pathogens [20]. Additionally, gut microbiota produces antimicrobial compounds, such as bacteriocins, that directly inhibit the growth of pathogenic bacteria [21]. Gut microbiota plays a vital role in the digestion and fermentation of dietary components, such as complex carbohydrates, proteins, and fibers, that are otherwise indigestible by the host [22]. Microbial fermentation in the gut produces short-chain fatty acids (SCFAs), vitamins (e.g., B and K vitamins), and other metabolites that are essential for host nutrition and health. The microbial fermentation process also produces SCFAs, such as acetate, propionate, and butyrate, as byproducts [23]. These SCFAs are absorbed through the rumen wall and serve as important sources of energy for the cow. Additionally, the rumen microbiota plays a role in synthesizing certain vitamins, such as B vitamins, which are essential for the cow’s metabolism and overall health. Ruminants, such as cattle and sheep, harbor a diverse microbial ecosystem in their rumen, which allows them to efficiently utilize plant materials. This microbial community helps in the digestion of lignocellulosic materials and non-protein nitrogen, reducing competition for human-edible foods [24]. The gut microbiota significantly influences feed conversion efficiency (FCE) in livestock animals. Microbial fermentation in the rumen of ruminants, for example, converts low-quality plant material into metabolizable energy for the host. Optimization of the rumen microbiota composition and activity can improve FCE, leading to reduced feed costs and environmental impacts. Methane emissions from ruminant livestock contribute to greenhouse gas emissions and energy loss in animals [25,26]. Strategies aimed at modulating the gut microbiota, such as the use of probiotics or dietary manipulation, can reduce methane production by altering microbial metabolism in the rumen.

## 3. Exploring the Gut Microbiota

The general composition of animal gut microbiota can be broadly categorized based on the types of microorganisms present. It is important to note that the specific composition can vary widely across different species, diets, and environments, but certain phyla are commonly represented among various animals [27,28]. Major bacterial phyla such as Bacillota, Bacteroidetes, and Proteobacteria, along with various genera like Ruminococcus, Prevotella, and Fibrobacter, populate different regions of the GI tract [29]. Bacillota is one of the most abundant bacterial phyla in many animals, including humans. Within the Bacillota phylum, specific taxa such as Clostridiales and Lactobacillales are particularly prominent. These taxa are well adapted to the intestinal environment and contribute significantly to the stability and complexity of the gut microbial community [30]. Clostridiales is an order of Gram-positive, anaerobic bacteria that includes various genera known for their diverse metabolic capabilities and roles in gut health. Many species within Clostridiales are involved in fermenting dietary fibers and producing beneficial metabolites such asbutyrate, bile acids and indolepropionic acid, phosphatidylcholine and phenolics, which contribute to intestinal health and host metabolism [31]. Lactobacillales, on the other hand, consists mainly of lactic acid bacteria (LAB) known for their ability to produce lactic acid as a metabolic byproduct. Lactic acid production helps to create an acidic environment in the gut, which can inhibit the growth of potentially harmful pathogens. Additionally, some Lactobacillales species have probiotic properties and are commonly used in commercial probiotic products to promote gut health [32].

Another dominant phylum, members of the Bacteroidetes phylum, such as Bacteroides and Prevotella genera, excel in the breakdown of complex molecules like proteins and carbohydrates. Proteobacteria is a diverse phylum that includes many different classes of bacteria, including some that are commensal (such as Escherichia) and some that can be pathogenic (such as Salmonella and Helicobacter) [33,34]. Actinobacteria is often less abundant than Bacillota and Bacteroidetes; this phylum includes beneficial genera such as Bifidobacterium, which is known for its role in maintaining gut health [35]. Verrucomicrobia is a less common group but can be significant in certain animals; for example, the genus Akkermansia has been studied for its role in maintaining gut barrier function and metabolic health in cats and dogs [36]. The high abundance of Proteobacteria in animals and fish reflects their advantages as facultative anaerobes in environments where oxygen availability fluctuates. Facultative anaerobic bacteria like Proteobacteria exhibit highly flexible metabolic properties, enabling them to adapt to diverse environmental conditions [37]. They are specialists in host association, representing major symbionts and pathogens in agriculture. While Proteobacteria are ubiquitous, they also display host-specific associations in certain microbiota. For example, in fish intestinal microbiota, Aeromonadaceae are predominant in freshwater fish, whereas Vibrionaceae dominate in marine fish [38]. In livestock animal microbiota, Enterobacteriaceae, Campylobacteriaceae, and Helicobacteraceae are major contributors, posing potential risks for foodborne diseases in humans. Bacillota, another dominant phylum in animals and fish, encompasses lactic acid bacteria and anaerobic fermentative bacteria [39]. Lactic acid bacteria are prevalent in oxic to microoxic regions like plant phyllosphere and fish mucosa, whereas anaerobic fermentative bacteria are common in anoxic environments like animal and fish intestines [32]. Bacteroidetes, which colonize animals and fish, include aerobic Flavobacteriaceae and anaerobic fermentative bacteria like Bacteroidaceae and Prevotellaceae. Flavobacteriaceae are adapted to toxic environments and can act as both pathogens and growth-promoting microbes [40]. Bacteroidaceae and Prevotellaceae are primary fermenters in animal and fish intestinal tracts, aiding in the breakdown of complex carbohydrates and undigested proteins. Notably, microbiota associated with animals and fish exhibit high diversity and can harbor up to 20 bacterial phyla; however, three phyla such as Proteobacteria, Bacillota, and Bacteroidetes tend to dominate bacterial communities across various hosts [41].

Members of the archaea domain are less abundant than bacteria but can play important roles in the gut ecosystem. For instance, Methanogens are archaea that produce methane as a byproduct of anaerobic digestion, and they are often found in the guts of ruminant animals like cows and sheep [41]. Meanwhile, eukaryote such as fungi, a mycobiome (fungal component of the microbiome) can include yeasts and molds. Candida and Saccharomyces are common genera found in some bovine animals such as cow, cattle, and buffalo [42]. Protists can be commensal or parasitic. Some protists are important for cellulose digestion in the guts of herbivores [43]. In addition, viruses such as bacteriophages, which are viruses that infect bacteria, are abundant in the gut and can significantly impact bacterial populations by causing bacterial cell lysis. Eukaryotic viruses capable of infecting a broad spectrum of animal hosts, including primates, birds, reptiles, and amphibians, represent a diverse array of families and genera. Notable examples encompass adenoviruses, herpesviruses, retroviruses, papillomaviruses, orthomyxoviruses, and paramyxoviruses. These viruses can induce various diseases ranging from respiratory infections to tumors, showcasing their significant impact on both animal and human health [44].

An example of a protozoa commonly found in the rumen of ruminant animals is Entodinium. Entodinium is a genus of ciliate protozoa characterized by its large size and complex morphology, making it well suited for the breakdown of ingested plant material and microbial protein within the rumen environment [45]. Entodinium protozoa possess specialized structures called cytostomes, which are used for ingesting feed particles and microorganisms. Within their cytoplasm, Entodinium harbors proteolytic enzymes that enable them to degrade proteins into smaller peptides and amino acids. This enzymatic activity allows Entodinium to efficiently utilize proteinaceous material as a nitrogen source for their own growth and metabolism [46]. Furthermore, Entodinium and other protozoa in the rumen contribute to intraluminal nitrogen recycling by breaking down microbial protein, particularly bacterial protein. This process releases ammonia, which is then utilized by other rumen microbes, such as bacteria, to synthesize microbial protein. The microbial protein synthesized by bacteria serves as a vital source of high-quality protein for the host animal, ultimately contributing to its overall protein nutrition and health [47]. Table 1 displays the general composition of animal gut microbiota.

## 4. Key Players in a Healthy Microbiome

The composition of the intestinal microbiota is strongly influenced by the ecological niches provided by the host organism, with environmental conditions such as temperature, pH, oxygen levels, and nutrient availability playing crucial roles [48]. These factors include not only the genetic makeup and physiological characteristics of the host but also environmental conditions that directly impact microbial growth and survival within the gut environment [49,50]. One key environmental factor is temperature, as microbial growth rates and metabolic activities are highly temperature dependent. The gut provides a relatively stable temperature range conducive to the growth of certain microbial species, influencing the diversity and abundance of gut microbiota. Fluctuations in temperature, whether due to external environmental factors or host physiological changes, can alter microbial composition [51].

The pH levels within the gut also play a critical role in shaping the intestinal microbiota. Different regions of the GI tract exhibit varying pH levels, creating distinct microenvironments that favor specific microbial species. For instance, the acidic environment of the stomach selects for acid-tolerant bacteria, while the more neutral pH of the small intestine and colon supports a different set of microbial communities [52]. Oxygen levels within the gut vary across different regions, with the small intestine being relatively oxygen-rich compared to the anaerobic conditions prevailing in the colon. This oxygen gradient influences the distribution of aerobic and anaerobic microorganisms along the length of the GI tract, ultimately shaping the composition of the gut microbiota [53]. Nutrient availability is another critical environmental factor influencing gut microbial composition. The gut provides a diverse array of nutrients derived from dietary intake and host secretions, serving as a rich substrate for microbial growth. Microbial species with specialized metabolic capabilities can thrive in niches where specific nutrients are abundant, leading to the establishment of unique microbial communities within different regions of the gut [54]. Overall, the ecological niches provided by the host organism, in conjunction with environmental conditions such as temperature, pH, oxygen levels, and nutrient availability, collectively determine the composition and diversity of the intestinal microbiota. Understanding the interplay between these factors is crucial for unraveling the complex dynamics of host–microbe interactions and their implications for host health and physiology [55].

However, the animal genetic makeup can make it susceptible to colonization by certain microbes. For example, genetic variations in mucosal barrier function or immune response can affect which microbes can establish themselves in the gut [56]. Diet is probably the most significant environmental factor, since diet directly influences which microbial species can survive and thrive in the gut. Habitat, the local environment, including soil, water, and available flora and fauna, provides a source of microbial species that can colonize the gut [57]. Psychological and physiological stressors can impact gut microbiota composition and function through the activation of the gut–brain axis and the release of stress hormones. Chronic stress can alter gut permeability, immune responses, and microbial diversity, contributing to gut dysbiosis and associated health issues [58,59]. Husbandry practices, including feeding regimens, hygiene protocols, and disease management strategies, influence gut microbiota composition and overall gut health in animals [60]. Optimal management practices that prioritize nutrition, sanitation, and stress reduction are essential for maintaining a healthy gut microbiota and maximizing animal productivity. Interrelations between different microbial species within the gut microbiota can shape community structure and function. Competition, cooperation, and cross-feeding interactions among microbes influence microbial diversity and metabolic activities in the gut ecosystem. Gut microbiota composition is often altered in response to infectious diseases, inflammatory conditions, or metabolic disorders. Pathogen invasion, immune activation, and tissue damage can disrupt microbial communities and impair gut barrier function, leading to further complications [61]. Lastly, exposure to antibiotics, either through medical treatment or environmental contamination, can dramatically alter the gut microbiome. The human body encounters environmental toxins primarily through the digestive tract and the respiratory system. These toxins undergo metabolic transformations by both human and microbial enzymes, with microbial reactions often differing from host metabolism [62]. For instance, while host enzymes typically oxidize and conjugate toxins for excretion, microbial enzymes predominantly perform reduction, hydrolysis, and demethylation reactions. The microbial metabolism of environmental chemicals, including heavy metals and endocrine disruptors, can influence health outcomes, potentially leading to dysbiosis and altered microbial transformation processes [63]. Enzymes such as azoreductases, esterases, methylases, and sulfatases are among those involved in microbial metabolism. Persistent chemicals from personal care products, such as triclocarban and triclosan, are pervasive and can impact the microbiome. Understanding these interactions is vital for assessing health risks associated with environmental exposures [64,65].

When populations of healthy gut organisms diminish or there is insufficient diversity in the microbiota, which refers to the collection of microorganisms inhabiting the body, various GIs and immune-related issues can arise. Disruptions in the composition and diversity of the intestinal microbiota, known as dysbiosis, can occur due to factors such as diet changes, antibiotic use, stress, and disease conditions. Dysbiosis in animals usually has been linked to various health issues, including GI disorders, metabolic diseases, and reduced production efficiency [66,67], potentially leading to small intestinal bacterial overgrowth (SIBO) and leaky gut syndrome. Research indicates that animals lacking rich microbial diversity or adequate colonies of friendly bacteria in their gut, or those experiencing imbalances in their microbiome characterized by a poor ratio of beneficial to harmful gut bacteria, are at heightened risk of developing a wide range of chronic diseases. These findings underscore the critical role of maintaining a healthy balance of gut microbiota in promoting overall health and preventing the onset of chronic conditions in animals [68,69]. Table 2 shows key factors influencing the composition and diversity of the intestinal microbiota in animals.

The gut microbiome of animals plays a crucial role in regulating biomolecules in biofluids. Bacteria such as Lactobacillales produce extracellular vesicles (EVs), and recent studies suggest that interactions between gut bacteria and host cells, especially epithelial and immune cells, may influence the production and release of EVs by host cells [69]. These EVs from animals may carry molecules influenced by or derived from the gut microbiota, such as bacterial components, metabolites, and signaling molecules. These interactions could indirectly impact the production or content of host-derived EVs in the gut. Investigating how gut bacteria influence EV biology could offer valuable insights into the mechanisms behind the health benefits of probiotics and modulating the gut microbiota. EVs can traverse the mucus layer, cross the epithelial barrier, and disseminate throughout the body, suggesting that EVs predominantly facilitate communication between different kingdoms in the gut [70].

## 5. EVs as Communication Mediators between Gut Microbiome and Host

Initially regarded as cellular waste, exosomes were first identified in 1981 by Trams et al. as exfoliated membrane vesicles containing ecto-enzymes [71]. Subsequent research by Pan and Johnstone in 1983 observed their release by maturing sheep reticulocytes. However, it was not until 1987 that they were termed “exosomes,” and their physiological significance was recognized in 1996 [72,73,74]. EVs are non-replicating membrane-bound entities produced by cells, playing diverse roles and reflecting the physiological states of their parent cells. They have emerged as potent mediators of intercellular communication, sparking renewed interest in their classification and potential applications in various fields. EVs have been isolated from various biological fluids, including plasma, serum, urine, saliva, bronchial secretions, breast milk, amniotic fluid, and seminal fluid. Some well-studied subtypes of EVs include ectosomes, a type of EV that emerge via direct budding or “shedding” from a cell’s plasma membrane. In the meantime, exosomes originating from intracellular budding are released by cells. Typically ranging from 40 to 120 nm in size, exosomes carry bioactive molecules and are secreted by various cell types in both physiological and pathological conditions. Another type of EV is apoptotic bodies, which are remnants of cells undergoing apoptosis, or programmed cell death. Oncosomes, larger EVs produced by cancer cells, neurons, and other cell types, bear a striking resemblance to cells themselves. Microsomes are small endoplasmic reticulum (ER)-derived EVs produced artificially during tissue homogenization, used for ER structure and function studies. Liposomes are phospholipid bilayer-delimited EVs used extensively in biocompatible drug delivery systems. Lastly, micelles are tiny lipid monolayer-delimited EVs enclosing a hydrophobic interior suitable for delivering fat-soluble drugs and other compounds [71,72,73,74,75].

EVs, particularly exosomes, play a crucial role in cell-to-cell communication, carrying a cargo of proteins, DNA, RNA, microRNAs (miRNAs), cytokines, metabolites, and lipids. While the exact mechanism of interaction between exosomes and target cells is not fully understood, specific molecules on exosomal membranes are thought to facilitate binding [76]. Additionally, some exosomes release their cargo outside the cell without direct interaction, possibly through molecules binding to cell receptors. Exosomes are involved in various physiological and pathological processes, including immune response, viral pathogenicity, pregnancy, cardiovascular diseases, central nervous system-related diseases, and cancer progression. Their diverse roles make them potential candidates for therapeutic and diagnostic applications [77]. Engineered exosomes can deliver therapeutic payloads, while exosome-based liquid biopsy has shown promise in diagnosing and prognosticating various diseases. Isolating exosomes is a current area of research, with several techniques proposed, each with its advantages and disadvantages. Techniques for evaluating the quality of harvested exosomes involves assessing their number, concentration, size, morphology, composition, and cargo. Techniques such as nanoparticle tracking analysis (NTA) and electron microscopy (EM) are commonly used to determine general features of EV samples. Flow cytometry, including bead-based detection methods and imaging flow cytometers, is applied for the quantification and membrane marker detection of EVs. However, detecting EVs with flow cytometers is challenging due to their small size, and efforts are ongoing to improve EV flow cytometry analysis and standardization [78,79].

Exosomal content includes proteins such as tetraspanins, ALIX, and TSG101, DNA, RNA including miRNAs, and various lipids like sphingomyelin, cholesterol, and ceramide [80]. These macromolecules play critical roles in inflammation, angiogenesis, immune response, cancer, and neurodegenerative diseases. The structure of exosomes and their cargo facilitate multicellular crosstalk, mediating cell signaling and intercellular transfer of biomolecules [81]. Payload carried by exosomes contribute to various cellular functions and have significant implications for human and veterinary medicine. EVs, like exosomes, released by both gut bacteria and eukaryotic cells in response to various stimuli, such as infection or stress, can carry inflammatory molecules, including pro-inflammatory cytokines and inflammation-associated RNAs [82]. These EVs may contribute to neuroinflammation, which has been implicated in the pathogenesis of various mental disorders, including depression, anxiety, bipolar disorder, and schizophrenia [83].

EVs serve as key mediators of intercellular communication, operating across various levels within and between organisms. Their significance in shaping immune system dynamics is particularly noteworthy, as they play essential roles in modulating both innate and adaptive immunity [84]. This interplay is crucial in contexts such as chronic inflammatory diseases and allergies, where immune responses are dysregulated. Moreover, EVs facilitate the transfer of information not only within an organism but also between organisms. For example, animal-derived products like milk contain EVs that can carry bioactive molecules, including nucleic acids and proteins, which can be transferred to recipient cells upon consumption. Studying this transfer of information through EVs sheds light on broader ecological and physiological implications of intercellular communication. Understanding the role of EV-mediated communication in immune regulation and disease pathogenesis is vital for developing novel therapeutic strategies and addressing societal challenges related to health and wellness. By unraveling the complexities of EV biology and their functions in intercellular communication, researchers can uncover new insights into immune system dynamics and potentially harness the therapeutic potential of EVs for various medical applications [85].

## 6. Microbiota EVs

Gut microbiota-derived EVs (MDEVs) can also influence host health by transporting molecules across barriers like the intestinal and brain barriers [86,87]. Dysregulation of the gut–brain axis has been implicated in the pathogenesis of various disorders, including irritable bowel syndrome (IBS), inflammatory bowel disease (IBD), obesity, anxiety, depression, and neurodegenerative diseases. IBD is a chronic condition of the GI tract that affects both dogs and cats. It is characterized by recurrent or chronic symptoms such as vomiting and/or diarrhea, with vomiting being the most common sign in cats with IBD [88]. Despite these symptoms, affected animals may appear otherwise normal; however, weight loss may occur in some cases. Animals with IBD typically have a normal or increased appetite. The exact cause of IBD is not well understood, but it is believed to involve an abnormal immune response in the bowel lining, leading to infiltration of inflammatory cells. This can disrupt the normal digestive and absorptive functions of the intestine and may result in thickening of the intestinal wall. While the precise underlying cause is often unknown, dietary sensitivities or reactions to bacterial proteins are commonly suspected triggers. There is growing interest in understanding the potential interplay between MDEVs and IBD. Recent studies have suggested that MDEVs could play a role in the pathogenesis of IBD through trigger immune responses via immunostimulatory molecules like lipopolysaccharides (LPS), peptidoglycans (PG), and microbial proteins, activating pattern recognition receptors (PRRs) on immune cells, leading to chronic inflammation. They disrupt gut epithelial barriers, leading to damage, apoptosis, and increased permeability, which in turn worsens inflammation. MDEVs may disrupt the integrity of the intestinal epithelial barrier, allowing the translocation of microbial antigens and inflammatory mediators into the mucosa. This breach in barrier function could exacerbate inflammation and contribute to the pathogenesis of IBD. They also disrupt immune tolerance mechanisms, promoting aberrant immune responses against commensal bacteria and self-antigens, worsening IBD symptoms [89,90].

The interplay between MDEVs and obesity represents a multifaceted relationship with profound implications for metabolic health. MDEVs can exert metabolic effects by carrying molecules that influence adipogenesis, insulin sensitivity, and inflammation, thereby contributing to the development of obesity-related metabolic dysfunction. Moreover, these vesicles play a role in shaping the composition of the gut microbiota, favoring microbial populations associated with increased energy harvest and adiposity. Additionally, MDEVs may trigger inflammatory responses in metabolic organs, perpetuating a chronic low-grade inflammatory state characteristic of obesity. Hormonal regulation related to appetite and energy balance may also be influenced by MDEVs, further exacerbating dysregulated energy homeostasis in obesity [91].

The gut microbiome has been increasingly recognized as a potential contributor to various neurological disorders among young animals, including myelin disorders and mitochondrial encephalopathies. While the direct mechanisms linking gut microbiota to these conditions are not fully understood, emerging research suggests several potential pathways through which gut dysbiosis could influence the pathogenesis and progression of these disorders. Conversely, alterations in the EVs in gut microbiota can influence disease progression by modulating neuroinflammation, neurotransmitter production, and gut–brain axis signaling. This bidirectional communication underscores the potential of gut microbiome-targeted interventions as therapeutic avenues for managing neurodegenerative diseases in animals [92]. Similarly, in mitochondrial encephalopathies affecting dogs, characterized by mitochondrial dysfunction leading to neurological symptoms, the gut microbiome may contribute to the disease pathogenesis. Mitochondrial function can be influenced by microbial metabolites, such as SCFAs, produced by gut bacteria. Dysbiosis-induced changes in SCFA levels or other microbial-derived metabolites could potentially impact mitochondrial function and contribute to the pathophysiology of mitochondrial encephalopathies [93,94].

The interplay between host social behavior and the gut microbiome represents a dynamic and reciprocal relationship that has garnered significant attention in recent research [95]. Transmission of gut microbiota, whereby both vertical transmission from mothers and horizontal transmission from the environment play crucial roles in the establishment of the gut microbiota in newborn animals. Social interactions, such as grooming, mating, and fecal consumption, can promote the horizontal transmission of gut microbiota among individuals within social groups. Social behavior can influence the composition and diversity of the gut microbiome, while the gut microbiome can also impact host behavior [96]. For example, dysbiosis induced by antibiotic treatment in mice was found to reduce the sexual attractiveness of females to males, highlighting the role of the MDEVs in shaping reproductive behavior [97]. The composition of the gut microbiome, influenced by social behavior, can affect host health and fitness. Sociable individuals were found to harbor a gut microbiota enriched with beneficial bacteria associated with anti-inflammatory properties, suggesting a link between sociability and host health [98]. The study of social behavior and the gut microbiome in wildlife populations has implications for conservation biology. By elucidating the mechanisms underlying these interactions, interventions targeting the gut microbiome may be developed to promote the health and resilience of endangered species. Table 3 summarizes how MDEVs influence various aspects of host health.

Recent studies indicate that diet can affect the composition and characteristics of gut microbial EVs. For instance, high-fat diets have been shown to alter the size and composition of EVs, affecting insulin resistance and glucose intolerance. The changes in gut microbial EVs due to dietary factors can have significant implications for host health. For example, they can influence brain function, metabolism, gut function, and immune responses. The dietary protein was found to influence the production of secretory IgA through gut microbial EVs, affecting gut function and immune responses [82,99]. Overall, the gut microbiome exerts a significant influence on the production, composition, and function of exosomes in animals. Understanding the complex interplay between the gut microbiome and exosomes may provide insights into the mechanisms underlying host–microbiome interactions and their impact on health and disease [100,101,102,103,104,105].

## 7. Therapeutic Approaches to Modulate Gut Microbiota

Various therapeutic approaches are utilized to modulate the GI tract microbiota, emphasizing their potential to enhance host health. Probiotics are living microorganisms naturally found in the GI tract, which have a beneficial impact on host health. They work by producing metabolites that promote the growth of beneficial bacteria, inhibit pathogenic bacteria, regulate pH, enhance mucus production, and improve intestinal epithelial cell function [106,107]. In livestock production, probiotics are commonly used to improve GI tract health, feed efficiency, and milk quality. They can also help prevent dysbiosis during stressful events like transportation. For instance, deoxynivalenol (DON), a common food-related mycotoxin, was found to disrupt the gut microbiota, trigger immune imbalance, and damage the intestinal barrier in mice [108,109]. However, administration of *Lactobacillus murinus* (*L. murinus*), or its EVs, reversed DON-induced growth retardation, immune disorders, and intestinal barrier imbalance. Mechanistically, *L. murinus* and its EVs modulated macrophage phenotype, shifting them from the pro-inflammatory M1 to the anti-inflammatory M2 phenotype. These findings suggest the therapeutic potential of probiotics, particularly *L. murinus* and its EVs, in mitigating DON-induced intestinal toxicity by modulating the gut microbiota, macrophage phenotype, and intestinal barrier function [110]. Other probiotics such as *Saccharomyces cerevisiae*, Lactobacillus, and Bifidobacterium also have been proven to improve the gut health by modulating the gut microbiota [111].

Prebiotics are substrates that bacteria in the GI tract utilize, promoting the growth of beneficial bacteria and conferring health benefits to the host [112]. Their relationship with gut microbiota-derived exosomes is an emerging area of research with significant implications for host health and performance. The substrates, including non-starch polysaccharides (NSP) or oligosaccharides, are indigestible by the host but fermentable by commensal GIT microbiota. Prebiotics can enhance weight gain, feed efficiency, and overall health in cattle. Examples include fructose oligosaccharides (FOS) and galactosyl-lactose (GL), which have been shown to reduce enteric issues and improve growth in calves [113]. FOS, the non-digestible sugars serve as food sources for beneficial bacteria in the large intestine of pets [114]. By fermenting FOS, these bacteria contribute to overall GI health, improve gut microbiome ecology, and enhance fecal quality. In livestock and poultry production, FOS, derived from lysogenic fructose, are utilized to control pathogenic bacteria, minimize fecal odors, and enhance growth performance. For instance, studies involving broiler chickens have demonstrated that dietary supplementation with FOS can lead to improved growth performance, bolster innate and acquired immune responses, and enhance the structure of the intestinal mucosa [115]. Overall, the relationship between prebiotics, gut microbiota, and gut microbiota-derived exosomes represents a complex network of interactions with implications for host health and performance. Further research is needed to elucidate the mechanisms underlying these interactions and to explore the potential therapeutic applications of targeting gut microbiota-derived exosomes in conjunction with prebiotic supplementation.

Gut microbial transplants involve the transfer of microbial populations from a healthy donor to a recipient experiencing dysbiosis [116]. In ruminant animals, ruminal fluid transplants (RFT) are commonly used to introduce rumen fluid from a healthy donor to a recipient. This approach can accelerate rumen fermentation, decrease dysbiosis, repair damage to ruminal epithelial cells, and improve starch digestibility. The impact of RFT on the rumen microbial composition and growth performance of yaks transitioning from natural pastures to house-feeding periods was investigated, whereby RFT significantly influenced rumen alpha diversity, with the RFT group exhibiting higher OTU numbers and diversity metrics. Analysis of rumen microbiota composition revealed differences between groups, with lower abundances of Bacteroidota, Proteobacteria, and Spirochaetes, and higher abundance of Bacillota in the RFT group [117]. Overall, these findings suggest that RFT improves yak growth performance and reshapes the rumen microbial community, offering insights into microbial transplantation in yaks and potential strategies for enhancing feed efficiency in the industry [118].

The potential benefits of early microbial intervention through fresh rumen microbiota transplantation (RMT) and sterile RMT in postpartum dairy cows were also investigated [119]. RMT was found to expedite the transition process of ruminal microbiota in postpartum dairy cows, but may not significantly impact dry matter intake or feed efficiency, indicating limited benefits in promoting postpartum recovery Interestingly, calves that received fecal matter transplants (FMT) exhibited higher relative abundance of Lactobacillus species and lower abundance of Clostridium and Bacteroides. The study underscores the importance of rigorous donor selection criteria, free from pathogens and previous disease or antibiotic treatment, when developing FMT products [120]. However, another study examined the effects of FMT from yaks, whereby it has increased the relative abundance of beneficial bacteria, enhanced microbial network complexity, and promoted essential metabolic and cellular processes in weaned calves. Overall, these findings suggest that FMT could be a valuable strategy for preventing weaning diarrhea and other intestinal diseases in ruminants [121].

Organic acids have gained attention as therapeutic agents for modulating the gut microbiota in animals due to their ability to influence microbial composition and activity. Organic acids, such as acetic acid, propionic acid, and butyric acid, exhibit antimicrobial properties against pathogenic bacteria in the GI tract. They can inhibit the growth of pathogens by lowering the pH of the gut environment, disrupting bacterial cell membranes, and interfering with microbial metabolism [122]. For instance, a 6% acetic acid solution can kill *Mycobacterium tuberculosis* after 30 min [123]. While organic acids can suppress the growth of pathogens, they also promote the proliferation of beneficial bacteria, such as Lactobacillus and Bifidobacterium species [124]. These beneficial bacteria contribute to gut health by producing SCFAs, enhancing nutrient absorption, and supporting the host immune system. Organic acids also contribute to the maintenance of gut health by modulating the composition and activity of the gut microbiota [35]. By promoting a balanced microbial community, organic acids can help prevent GI disorders, such as diarrhea and dysbiosis, in animals. Organic acids have been used to reduce the colonization of enteric pathogens, such as Salmonella and *Escherichia coli*, in the GI tract of animals [125]. By creating an unfavourable environment for pathogen growth and enhancing the competitive exclusion of pathogens by beneficial bacteria, organic acids can help prevent and control enteric infections.

In addition to prebiotics and probiotics, there is a growing interest in phage therapy in both agricultural and clinical fields. This therapy involves delivering bacteriophages to their targeted sites and serves as a supplemental treatment to enhance gut microbiota [126]. A study that investigated the effects of dietary supplementation with freeze-dried *Escherichia coli* phage cocktail, commercial probiotics, and their combination on the growth performance and gut microbiota diversity of broiler chickens demonstrated that supplementing chickens with a combination of phage cocktail and probiotics may have positively influenced growth performance and modulated the gut microbiota [127]. The group supplemented with a specific dosage of the phage cocktail exhibited significantly better growth performance compared to the control group. Interestingly, the presence of SCFA producers, known for their roles in facilitating carbohydrate breakdown and SCFA production, was significantly higher in the phage-supplemented chicken groups. Furthermore, microbial predicted metagenome analysis indicated upregulation of genes related to nutrient digestion, absorption, and energy production in the phage-supplemented groups. This suggests that supplementation with phages and probiotics modulates the gut microbiota, leading to enhanced growth performance [128]. SalmoFree, a salmonella phage treatment, demonstrated a beneficial impact on broiler chickens, with notable effects observed in the core microbiome [129]. Specifically, during the later stages of the production cycle, the core microbiome comprised species essential for microbiota adaptation, suggesting the efficacy of SalmoFree in promoting microbiome stability and resilience in broiler chickens. Specifically, species such as Eisenbergiella and Lachnoclostridium, which are important for degrading complex polysaccharides and producing SCFAs, were identified. Importantly, it led to a significant reduction in Campylobacter, a common pathogen in poultry, which is a positive outcome in terms of food safety. Additionally, there was an increase in Butyricimonas, Helicobacter, and Rikenellaceae, which are known inhabitants of the chicken gut with both negative and positive effects on health and metabolism. Hence, further research is essential for the development and implementation of large-scale phage therapy technologies in poultry production systems, ultimately contributing to improved animal health and food safety [130,131]. Figure 1 displays the summary of the gut microbiome and EVs in host animals.

## 8. Conclusions and Future Direction

In conclusion, this study underscores the multifaceted interplay between the gut microbiota, environmental factors, and host physiology in animals, highlighting the critical importance of maintaining a balanced microbial community for optimal health and well-being [132]. Therapeutic interventions such as probiotics, prebiotics, gut microbial transplants, organic acids, and phage therapy offer promising avenues for modulating the gut microbiome EVs and mitigating dysbiosis-related health issues in animals. However, further research is needed to elucidate the underlying mechanisms of action, explore personalized approaches to microbiome engineering, consider broader ecological implications, and translate findings into clinical applications. By addressing these research directions, we can unlock the full potential of gut microbiota modulation to improve animal health outcomes, enhance productivity, and promote sustainability in animal agriculture, ultimately benefiting both animal and human populations.

Looking ahead, future research should focus on advancing our understanding of microbiome-host interactions, developing precision microbiome engineering strategies, establishing regulatory frameworks, and translating scientific discoveries into practical applications in animal agriculture. By investigating the underlying mechanisms of therapeutic interventions, researchers can uncover novel targets for microbiota modulation and refine existing approaches for optimal efficacy. Additionally, personalized approaches to microbiome engineering could revolutionize veterinary medicine by tailoring interventions to individual animal characteristics and health status. Regulatory frameworks must be established to ensure the safe and responsible use of microbiome-based therapies in animal production systems, balancing the benefits of microbiota modulation with potential risks to animal and human health. By addressing these challenges and opportunities, we can harness the full potential of gut microbiota modulation to improve animal welfare, enhance agricultural sustainability, and advance our understanding of microbiome biology.

## Figures and Tables

**Figure 1 ijms-25-04024-f001:**
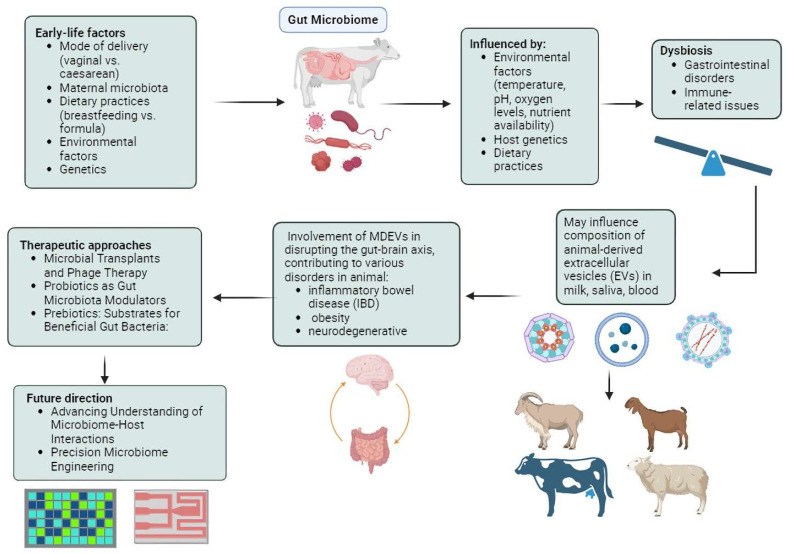
Environmental factors such as temperature, pH, oxygen levels, and nutrient availability profoundly shape the composition and diversity of the intestinal microbiota in animals, influencing their health and metabolism. Gut microbiota-derived extracellular vesicles (EVs), particularly exosomes, play crucial roles in intercellular communication, impacting immune response, neuroinflammation, and metabolic dysfunction. Therapeutic interventions like probiotics, prebiotics, microbial transplants, organic acids, and phage therapy offer promising avenues for modulating the gut microbiota, improving growth performance, health outcomes, and disease resilience in animals.

**Table 1 ijms-25-04024-t001:** General composition of animal gut microbiota.

Domain/Phylum	Description	Examples	Importance
Bacteria	Most abundant	Bacillota (Clostridiales, Lactobacillales), Bacteroidetes (Bacteroides, Prevotella), Proteobacteria (Escherichia, Salmonella), Actinobacteria (Bifidobacterium)	Digestion, nutrient fermentation, immune modulation, pathogen inhibition
Archaea	Less abundant	Methanogens	Methane production in ruminants
Eukarya (Mycobiota)	Fungi (yeasts, molds)	Candida, Saccharomyces	Gut health
Eukarya (Protista)	Commen-sal or parasitic	Entodinium (ciliate protozoa)	Cellulose digestion (herbivores)
Viruses	Abundant	Bacteriophages, eukaryotic viruses	Regulate bacterial populations

**Table 2 ijms-25-04024-t002:** Factors influencing gut microbiota composition in animals.

Factor	Description	Impact
Host Ecology	Temperature, pH, Oxygen levels, Nutrient availability	Shapes microbial growth and survival
Host Genetics	Mucosal barrier function, Immune response	Influences susceptibility to specific microbes
Diet	Directly influences microbial survival and growth	Most significant environmental factor
Habitat	Local environment (soil, water, flora, fauna)	Provides source of colonizing microbes
Stress	Gut–brain axis activation, Stress hormone release	Alters gut permeability, immune response, and diversity
Husbandry Practices	Feeding regimens, Hygiene protocols, Disease management	Influences gut microbiota composition and overall gut health
Microbial Interactions	Competition, Cooperation, Cross-feeding	Shapes community structure and function
Disease	Pathogen invasion, Immune activation, Tissue damage	Disrupts microbial communities and impairs gut barrier function
Antibiotics	Medical treatment or environmental contamination	Dramatically alters gut microbiome
Environmental Toxins	Metabolism by gut microbes	May lead to dysbiosis and altered microbial transformation processes

**Table 3 ijms-25-04024-t003:** MDEVs influence various aspects of host health.

Host System	MDEV Effects	Potential Mechanisms	References
Gut–Brain Axis (IBD)	Disrupt gut barrier, worsen inflammation	Activate immune responses, disrupt barrier integrity, disrupt immune tolerance	[86,87]
Metabolic Health (Obesity)	Influence adipogenesis, insulin sensitivity, inflammation	Carry molecules affecting adipogenesis, etc., shape gut microbiota, trigger inflammation	[89,90]
Nervous System (Neurological Disorders)	Modulate neuroinflammation, neurotransmitter production	Modulate gut–brain axis signaling, influence SCFA levels	[91]
Behavior (Social Behavior)	Shape reproductive behavior, influence sociability	Horizontal transmission of gut microbiota, influence social behavior via metabolites	[92,93,94]

## Data Availability

Data are contained within the article.

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
