# Peer review of "The Profound Influence of Gut Microbiome and Extracellular Vesicles on Animal Health and Disease"

_ijms, 2024, doi:10.3390/ijms25074024_

Round 1

Reviewer 1 Report

Comments and Suggestions for Authors

The manuscript provides a comprehensive examination of the intricate relationship between the animal gut microbiota and host health, offering valuable insights into this complex field. The authors adeptly discuss various factors influencing microbiota composition, including host ecology, genetic variability, dietary habits, and environmental stressors, thus presenting a thorough overview of the subject matter. Particularly compelling is the analysis of dysbiosis and its potential impact on gastrointestinal and immune function, highlighting the significance of microbial equilibrium in maintaining animal well-being and productivity. Moreover, the incorporation of innovative therapeutic approaches such as phage therapy underscores the manuscript's relevance in addressing contemporary challenges in microbiota research. Overall, this manuscript represents a significant contribution to the field, although some revisions are needed to enhance clarity and coherence.

Main Flaws:

Throughout the manuscript, several arguments lack adequate citations, undermining the strength of the claims. For instance, references are missing in various sections (Lines 294-295, 451-452, 456-457, 476-477, 491-499, 523-527, 537-542, 556-558).

Minor Points:

1. In line 66, there's a misunderstanding regarding the distinction between microbes and microbiome. Microbiome refers to the collective genomes of microorganisms in an environment, while microbiota typically refers to the microorganisms themselves.

2. Line 71-73 asserts that commensal microbiota is crucial for gut structure and morphology. However, the claim may overstate the impact, considering germ-free animals do not necessarily exhibit significant structural or morphological issues.

3. In line 95-97, SCFAs and VFAs are synonymous terms; it's recommended to use only one term for consistency.

4. Line 117 refers to Firmicutes, which is now classified as Bacillota.

5. Line 223 introduces a new paragraph abruptly; it's advisable to maintain continuity with the preceding paragraph.

6. The resolution of Figure 1 is insufficient and should be improved for better clarity and interpretation.

Author Response

Dear Reviewer 1,

Greetings!

We are pleased to submit our revised manuscript titled “The profound influence of gut microbiome and extracellular vesicles on animal health and disease” for publication in your esteemed journal. We have highlighted the changes that was made in the manuscript according to the recommendation from the reviewer using point-by-point format.

Reply: Thank you for your recommendation.

  1. The manuscript provides a comprehensive examination of the intricate relationship between the animal gut microbiota and host health, offering valuable insights into this complex field. The authors adeptly discuss various factors influencing microbiota composition, including host ecology, genetic variability, dietary habits, and environmental stressors, thus presenting a thorough overview of the subject matter. Particularly compelling is the analysis of dysbiosis and its potential impact on gastrointestinal and immune function, highlighting the significance of microbial equilibrium in maintaining animal well-being and productivity. Moreover, the incorporation of innovative therapeutic approaches such as phage therapy underscores the manuscript's relevance in addressing contemporary challenges in microbiota research. Overall, this manuscript represents a significant contribution to the field, although some revisions are needed to enhance clarity and coherence.

Reply: Thank you for the recommendation. We have now revised the manuscript accordingly.

  1. Throughout the manuscript, several arguments lack adequate citations, undermining the strength of the claims. For instance, references are missing in various sections (Lines 294-295, 451-452, 456-457, 476-477, 491-499, 523-527, 537-542, 556-558).

Reply: Thank you for your suggestion. We have now cited relevant papers that contribute significantly to our understanding of the subject matter.

  1. In line 66, there's a misunderstanding regarding the distinction between microbes and microbiome. Microbiome refers to the collective genomes of microorganisms in an environment, while microbiota typically refers to the microorganisms themselves.

Reply: Thank you for the suggestion. We have changed from microbiome to microbiota. (highlighted in yellow)

  1. Line 71-73 asserts that commensal microbiota is crucial for gut structure and morphology. However, the claim may overstate the impact, considering germ-free animals do not necessarily exhibit significant structural or morphological issues.

Reply: Thank you for the suggestion. We have added in healthy animal (highlighted in yellow)

  1. In line 95-97, SCFAs and VFAs are synonymous terms; it's recommended to use only one term for consistency.

Reply: Thank you for the suggestion. We have now revised it.

  1. Line 117 refers to Firmicutes, which is now classified as Bacillota.

Reply: Thank you for the suggestion. We have now revised it.

  1. Line 223 introduces a new paragraph abruptly; it's advisable to maintain continuity with the preceding paragraph.

Reply: Thank you for the suggestion. We have revised it accordingly.

  1. The resolution of Figure 1 is insufficient and should be improved for better clarity and interpretation.

Reply: Thank you for the suggestion. We have revised it accordingly

Thank you for the opportunity to further explain our research hence we request you to consider the manuscript for publication in the esteemed journal and oblige.

Thank you.

Sincerely,

Barathan Muttiah

Reviewer 2 Report

Comments and Suggestions for Authors

Muttiah et al. present a review on the gut microbiome and how it’s EV production  can have a profound effect on animal health and disease. Overall, the review is very well written and comprehensive, providing a unique and insightful addition to animal health (both human and veterinarian). I have no major concerns and the author’s should be commended.

One minor thought that I had was triggered by the use of the word “performance”. In this context, performance is typically described in the review as growth performance. I know there’s some human literature that describes gut health on “performance” in terms of strength or skeletal muscle function. Is there any corresponding data in animals? It could be a further unique spin on the review but is not necessarily needed.

Author Response

Thank you for the recommendation. I have attached a letter for your query. 
